# Are Faecal Microbiota Analyses on Species-Level Suitable Clinical Biomarkers? A Pilot Study in Subjects with Morbid Obesity

**DOI:** 10.3390/microorganisms9030664

**Published:** 2021-03-23

**Authors:** Per G. Farup, Maria G. Maseng

**Affiliations:** 1Department of Research, Innlandet Hospital Trust, N-2381 Brumunddal, Norway; 2Department of Clinical and Molecular Medicine, Faculty of Medicine and Health Sciences, Norwegian University of Science and Technology, N-7491 Trondheim, Norway; 3Bio-Me, Gaustadalléen 21, N-0349 Oslo, Norway; maria@bio-me.com

**Keywords:** faecal microbiota, obesity, irritable bowel syndrome, biomarker, shotgun analyses, the HUNT study

## Abstract

Background: An abnormal faecal microbiota could be a causal factor for disease. This study evaluated a new method for faecal microbiota analysis in subjects with obesity and irritable bowel syndrome. Methods: The study had a matched case-control design. Forty-six subjects with morbid obesity (defined as BMI > 40 or >35 kg/m^2^ with obesity-related complications) of whom 23 had irritable bowel syndrome (IBS), were compared with 46 healthy volunteers. The faecal microbiota was analysed with Precision Microbiome Profiling (PMP™) which quantified 104 bacteria species. The primary aim was comparisons between the cases and controls. Results: Two men and 44 women with a mean age of 43.6 years were included in each of the groups; BMI in the groups was (mean and SD) 41.9 (3.5) and 22.5 (1.5) kg/m^2^, respectively. Seventeen bacterial species showed statistically significant differences between the groups after adjusting for multiple testing. In a post hoc analysis, the sensitivity and specificity were 78%. Alpha diversity was lower in the group with obesity. In subjects with morbid obesity, no clinically significant differences were seen between subjects with and without IBS or from before to six months after bariatric surgery. Conclusions: The results encourage further evaluation of the new microbiome profiling tool.

## 1. Introduction

The prevalence of obesity has tripled since 1975 and constitutes a considerable health problem worldwide due to the high risk of noncommunicable diseases and increased mortality [1]. The disease is due to an imbalance in the intake, uptake, and expenditure of nutrients. The mechanisms and pathophysiology of obesity and the microbiota-associated metabolic changes are complex and incompletely understood [2,3,4]. Studies with various methods have shown associations between obesity and the gut microbiota, but not valid and congruent characteristics of the abnormal faecal microbiota (dysbiosis) [2,3,5]. Attempts to restore a normal bacterial composition for the prevention or treatment of obesity have not so far reached established clinical practice [5,6]. Irritable bowel syndrome (IBS), a common comorbidity in subjects with obesity, has also been associated with faecal dysbiosis [7,8,9].

Different analysis methods and different taxonomic levels could explain the microbiota analyses’ diverging results in subjects with obesity and IBS. The field of microbiome analysis calls for standardisation and higher precision [10]. In subjects with obesity, analyses on phylum level have shown a high Firmicutes/Bacteroidetes ratio in several studies. Analyses on deeper taxonomic levels, including species and even strains, might improve the differentiation between health and disease and understanding of the microbe functions and roles [2,3,5,11]. The most commonly used method for microorganism identification is amplicon sequencing of the 16S rRNA gene, where the prokaryotic ribosomal RNA small subunit is targeted. This gene contains nine hypervariable regions (V1–V9) of varying conservation flanked by highly conserved regions. Sequences are highly homologous between species within the same genus, and it is therefore challenging to obtain species resolution by 16S rRNA sequencing [12,13].

Next generation methods like metagenomics shotgun sequencing allow a detailed description of the microbiota but are resource-demanding [14]. A new, commercially available, less demanding method is Precision Microbiome Profiling (PMP™) which allows direct quantification of commonly occurring bacterial species through targeted qPCR assays [15,16]. New, valid, reliable, and readily available microbiota analyses are in demand.

This pilot study’s paramount aim was to evaluate the new method’s diagnostic properties in subjects with morbid obesity and IBS. The primary aim was to compare the faecal microbiota in subjects with morbid obesity and healthy volunteers and secondary to compare the microbiota in subjects with morbid obesity with and without IBS and before and after bariatric surgery.

## 2. Materials and Methods 

### 2.1. Study Design and Participants

This study used previously collected data and faecal samples from two studies. The first dataset was from a study of subjects with morbid obesity treated at Innlandet Hospital Trust, Gjøvik, Norway [17,18,19,20]. Data from 46 subjects, 23 with and 23 without IBS, were arbitrarily selected; 19 had a follow-up visit and provided a faecal sample six months after bariatric surgery.

The second dataset was from the HUNT4 Survey, a population-based survey in Norway [21]. The Trøndelag Health Study (HUNT) is a collaboration between HUNT Research Centre (Faculty of Medicine and Health Sciences, Norwegian University of Science and Technology NTNU), Trøndelag County Council, Central Norway Regional Health Authority, and the Norwegian Institute of Public Health. Forty-six healthy subjects matched for age and sex with the subjects with morbid obesity were used as controls.

The study had a matched case-control design with comparisons between 46 subjects with and 46 without obesity, a non-matched case-control design of subjects with and without IBS, and a prospective cohort design of 19 subjects with morbid obesity from inclusion to six months after surgery.

### 2.2. Inclusion Criteria

The study at Innlandet Hospital Trust included subjects 18–65 years of age with morbid obesity (defined as BMI > 40 kg/m^2^ or >35 kg/m^2^ with obesity-related complications) without previous major gastrointestinal surgery, organic gastrointestinal disorders, alcohol and drug abuse, major psychiatric disorders, or severe somatic disorders not related to obesity.

The HUNT4 Survey included more than 56,000 adult inhabitants in the northern part of Trøndelag County, Norway. Forty-six age- and gender-matched subjects with BMI within the normal range and without previous or present morbidity were selected as healthy controls.

### 2.3. Interventions

After inclusion, the subjects with morbid obesity started with a six-month conservative weight loss intervention. During this period, they had regular individual and group meetings with nurses, nutritionists, doctors, and psychologists, receiving dietary advice, physical activity programs, and information about the operation and all consequences. In the last three weeks of the conservative treatment period, just before surgery, the participants followed a strict “crispbread diet” or an alternative meal replacement powder diet with 4200 kJ [17]. There was a follow-up six months after bariatric surgery (either Roux-en-Y gastric bypass or gastric sleeve) [22,23]. The regimen with two subsequent interventions was according to national and international recommendations. Faecal samples were collected at inclusion and the follow-up visit. Previous publications give more details [17,18,19,20]. 

The participants in the HUNT4 Survey filled in extensive questionnaires about somatic or psychiatric disorders. They had a physical examination, and blood, urine, and faecal samples were collected. 

### 2.4. Variables 

The study used the following subset of data from the previous studies judged as relevant for changes in the faecal microbiota:Demographic and anthropometric data: age (years), gender (male/female), height (m), body weight (kg), and body mass index (BMI; kg/m^2^).Dietary habits: The diet was assessed with a validated food frequency questionnaire, and the daily intake was based on the Norwegian food composition table [24,25]. Use of non-nutritive sweeteners (NNS) was calculated, one unit of NNS being a 100 mL beverage with NNS or two tablets/teaspoons of NNS.Irritable bowel syndrome (IBS) was diagnosed according to the Rome III criteria with the subgroups diarrhoea-predominant (IBS-D), constipation-predominant (IBS-C) and mixed (IBS-M) [26].Morbidity and use of drugs: diabetes and use of metformin (yes/no).Blood tests: C-reactive protein (CRP, normal range < 3.0 mg/L; a marker of inflammation), and s-zonulin (normal range < 38 ng/mL; a marker of inflammation and gastrointestinal permeability).Faecal samples from the subjects with obesity were collected at inclusion and six months after surgery. The samples were mixed with stool transport and recovery buffer (Roche, Basel, Switzerland) in a 1:3 ratio by vortexing. All samples were pulse centrifuged, and 600 µL was transferred to a 96-well Lysing Matrix E rack (MP Biomedicals Inc., Santa Ana, CA, USA). Samples were mechanically lysed twice at 1800 rpm, 40 s on 40 s rest, in a FastPrep-96™ (MP Biomedicals Inc.). Lysed samples were centrifuged (5 min, 1300× g, PlateSpin II centrifuge, Kubota, Tokyo, Japan), and 250 µL was incubated at 65°C for 15 min with 250 µL mag^TM^ maxi kit lysis buffer BLM (prod. no 40430) (LGC Genomics GmbH, Berlin, Germany) and 20 µL mag^TM^ maxi kit protease (LGC Genomics GmbH, Berlin, Germany). A 400 µL aliquot of each protease-treated faecal sample was used to extract total genomic DNA according to mag™ maxi kit instructions (LGC Genomics, Berlin, Germany), adjusted for a MagMAX™ express 96 DNA extraction robot (Life Technologies, Waltham, MA, USA) [27]. The DNA was further handled for the samples from the healthy controls as described below.Faecal samples from the healthy controls were collected on Bio-Me filter cards organised by HUNT4. Three 6 mm discs were punched out from each sample filter card, and microbial DNA was extracted using a Microbiome MagMAX Ultra kit (Thermo Fisher Scientific, Waltham, MA, USA) [28] essentially following the manufacturer’s recommendations on KingFisher™ Flex (ThermoFisher Scientific). The bacterial cell wall was disrupted using a VWR Star Beater at maximum settings for 2 min. Purified DNA was eluted in 200 µL MagMAX Elution Buffer, and DNA was quantified using PicoGreen and an F200 Infinite plate reader (Tecan).The sample microbiome DNA from obese and healthy subjects was analysed using Precision Microbiome Profiling (PMP™) qPCR panels for direct quantification of 104 target bacterial species on the QuantStudio™ 12K qPCR platform (Thermo Fisher Scientific). Liquid handling steps were automated and performed using epMotion™ 5700 (Eppendorf) and Accufill™ system (ThermoFisher Scientific). Absolute quantification of the number of genomic copies per µL for each bacterial taxon was interpolated from standard curves derived from quantified reference isolates (see Appendix A). The relative abundance (%) is the total number of copies for a given target divided by the sum of copies for all 104 targets. Results were provided for both absolute quantification and relative abundance of each bacterial taxon. The relative abundance (%) of the 104 target bacteria was used in this study.

### 2.5. Statistics

Comparisons between groups were performed with the Mann–Whitney U-test, within groups with the Wilcoxon signed rank test, and correlations with Spearman’s rho using IBM SPSS Statistics for Windows, version 26.0 (IBM Corp., Armonk, NY, USA). Adjusting for multiple testing was performed with false discovery rate ad modum Benjamini and Hochberg in R-studio version 1.2.5033 and reported as *q*-values. *p*- and *q*-values < 0.05 were judged as statistically significant. No formal sample size was calculated in this pilot study. The best species for separating healthy and morbidly obese subjects and comparing the alpha diversity in selected groups were used in post hoc analyses.

### 2.6. Ethics

The study was approved by the Regional Committee for Medical and Health Research Ethics South-East Norway (references 2012/966 and 2019/43353) and conducted in accordance with the Declaration of Helsinki. All participants gave written informed consent before inclusion in the study. 

## 3. Results

### 3.1. The Participants

Table 1 gives the participant characteristics in the groups with morbid obesity at inclusion and six months after surgery and healthy volunteers. The 19 participants 6 months after surgery were part of the 46 participants with morbid obesity at inclusion. In one participant, the subgroup classification of IBS was not possible.

### 3.2. Morbid Obesity versus Normal Weight

Out of 104 bacteria, the relative abundance of 28 bacteria (27%) was significantly different in subjects with and without morbid obesity. After adjusting for multiple testing, 17 (16%) were statistically significant with *q*-values < 0.05, of which six *q*-values were <0.001. Table 2 gives the details. The proportion of all bacteria with significant differences between the groups was lower in subjects with obesity.

The median of the sum of the relative abundance of the 17 bacteria showing differences between subjects with morbid obesity and normal weight was 8.76% and 25.18% respectively (*p* = 0.001). Figure 1a shows the relative abundance in the two groups. 

In post hoc analyses, the best separation between morbidly obese and healthy subjects was the sum of eight species (*Akkermansia muciniphila*, *Anaerobutyricum hallii*, *Anaerostipes hadrus*, *Bifidobacterium adolescentis*, *Blautia wexlerae*, *Faecalibacterium prausnitzii*, *Methanobrevibacter smithii*, and *Ruminococcus bromii*). These species were selected based on the largest differences between the subjects with morbid obesity and normal weight. The median of the sum of the relative abundance of these eight bacteria in subjects with morbid obesity and normal weight was 3.64% and 11.89% respectively (*p* < 0.001). Figure 1b shows the relative abundance in the two groups. The eight species’ ability to separate subjects with obesity and healthy controls is shown in a receiver operating characteristic (ROC) curve (Figure 1c). The AUC is 0.83 (95% CI 0.75: 0.92; *p* < 0.001). With a cut-off value < 7 indicating obesity, both the sensitivity and specificity are 0.78.

Three out of four with diabetes in the obesity group used metformin. In the users of metformin, the relative abundance (median) of the 17 bacteria was lower than in the non-users, 2.3% and 9.3% respectively (*p* = 0.004), and the relative abundance of the eight bacteria was 1.1% and 3.8% respectively (*p* = 0.03). In the only subject with diabetes not using metformin, the relative abundance of the bacterial groups was close to the median values in the obesity group. The eight bacterial groups’ diagnostic properties for the diagnosis of obesity were only marginally changed after excluding three subjects using metformin (AUC = 0.82; 95% CI: 0.74–0.91; *p* < 0.001).

Among the 17 bacteria showing differences between subjects with obesity and healthy controls, 11 were in the phylum Firmicutes and one in Bacteroidetes. The relative abundance of all bacteria was lower in subjects with obesity than in those with normal weight. 

### 3.3. Morbid Obesity: Before and after Treatment

There were significant changes in the levels of 14 out of 104 bacteria (13%) from before to after surgery. After correcting for multiple testing, the only statistically significant change was an increase in *Alistipes shahii* from 0.5% to 3.3% (median values) (*p* = 0.001; *q*-value = 0.018). The relative abundance of the group with 17 bacteria associated with obesity increased from 8.76% to 10.13% (*p* = 0.81) from before to after surgery, respectively. After surgery, the relative abundance was independent of the type of surgery. 

### 3.4. Morbid Obesity: Irritable Bowel Syndrome 

The levels of three bacteria out of 104 were significantly different in subjects with and without IBS; none were statistically significant after correcting for multiple testing. The sample size was too small for differentiation between the IBS subtypes. IBS was not associated with the relative abundance of either the group with 17 or 8 bacteria.

### 3.5. Morbid Obesity: Other Variables 

Four participants (9%) had diabetes. Six bacteria showed statistically significant differences in participants with and without diabetes; none was significant after adjusting for multiple testing. In three participants (7%) using metformin, five bacteria were significantly different in users and non-users, none was significant after adjusting for multiple testing.

Eleven bacteria were significantly associated with NNS intake, three bacteria with CRP, and three bacteria with zonulin; none were significant after adjusting for multiple testing. CRP, use of NNS or zonulin were also not associated with the relative abundance of the groups with 17 and 8 bacteria.

### 3.6. Alpha Diversity

Alpha diversity was analysed in post hoc analyses. The diversity was significantly lower in subjects with obesity than in healthy controls and increased six months after bariatric surgery. IBS and type of surgery were not associated with alpha diversity. Table 3 gives the details.

## 4. Discussion

The primary aim was to compare the microbiota in subjects with morbid obesity and normal weight. The relative abundance of 17 out of 104 selected species (16%) showed significant differences between the groups after correcting for multiple testing; seven of the adjusted *p*-values (*q*-values) were ≤0.001. This finding demonstrates the new method’s potential for the characterisation of the microbiota in subjects with obesity. 

Because the analytical methods vary and the analyses are on different taxonomic levels, comparisons with other studies are difficult. Differences between subjects with obesity and normal weight have been demonstrated on all taxonomic levels from phylum to strain [3,12,29,30,31,32,33]. Since the first report 15 years ago, a common finding on phylum level has been an increased Firmicutes/Bacteroidetes ratio in subjects with obesity [2,5,34]. Studies showing a normal Firmicutes/Bacteroidetes ratio in obese subjects report abnormalities on lower taxonomic levels [29,33]. The optimal taxonomic level for the characterisation of disease-related changes in the microbiota has not been clarified. Results from next generation sequencing and the finding that different *Lactobacillus* strains have opposite effects indicate that analyses on low taxonomic levels might be advantageous [14,31].

As used in this study, the species level analyses might show up as appropriate for clinical purposes. The relative number of all the statistically significant species showing differences between the groups was lower in the obese group than in the healthy group. Eleven of the 17 species were in the phylum Firmicutes and only one in Bacteroidetes, indicating a low Firmicutes/Bacteroidetes ratio. The analyses included only 104 out of more than 1000 different bacterial species and only a minority of all Firmicutes species, which might explain the apparent deviation from a high Firmicutes/Bacteroidetes ratio often reported in subjects with obesity [35]. 

Gupta et al. published a Gut Microbiome Health Index (GMHI) test based on 50 bacterial species that predicted the presence/absence of 12 diseases/disorders with an overall accuracy of 73.7% [30]. The GMHI showed significant differences between subjects with normal weight and overweight and obesity, but obesity was not the disease with the most marked difference from healthy subjects. Meijnikman et al. presented a test based on 52 bacterial species that predicted obesity with an AUC of 0.82 [11]. These two studies’ results are in the same order of magnitude as the results of the data-driven post hoc analyses (the ROC curve) in this study. The most predictive species in the study by Meijnikman et al. were not congruent with this study [11]. A test distinguishing all diseases from healthy subjects is tempting. However, the type of dysbiosis is probably disease-specific, and a disease-specific test will have the best diagnostic properties [19]. In all, the new method tested in this study showed good properties as a tool for characterising the faecal microbiota in subjects with morbid obesity.

The analyses did not show notable taxonomic differences from before to six months after bariatric surgery. There was a statistically significant change in only one bacterium (*Alistipes shahii*) after correcting for multiple testing. There was no significant change in the relative quantity of the 17 bacteria characterising morbid obesity. The genus *Alistipes*, which consists of 13 species, has been ascribed both health-promoting and deleterious effects [36]. *Alistipes shahii* reduces inflammation and has favourable effects in subjects with liver fibrosis and cardiovascular disease [36]. The microbial changes were not associated with the type of bariatric surgery. Most studies have shown alterations in the faecal microbiota after bariatric surgery, both in man and mouse, but not all have shown differences related to the operation type [37,38,39,40]. The weight-loss following bariatric surgery has partly been attributed to the microbiota changes since transplantation of faeces from operated to non-operated obese mice has induced weight loss [40]. Whether bariatric surgery normalises the microbiota is still uncertain [41].

IBS is a common and unexplained disorder characterised by abdominal pain and varying stool frequency and consistency. The aetiology and pathophysiology are unknown. The microbiota has been pointed at as the mediator of the disorder. Several studies have compared the faecal microbiota in subjects with and without IBS and the different subgroups of IBS. Although the results are contradictory, IBS seems to have a distinct microbiome pattern independent of subgroup [7,8,9]. This study showed no difference between subjects with and without IBS, possibly because the dysbiosis in subjects with morbid obesity camouflages the effect of IBS. No analyses were performed on either the IBS subgroups or bile acid malabsorption.

The microbiota in subjects with morbid obesity is influenced by the high prevalence of comorbidity, their often unusual dietary habits, and use of drugs. Diabetes, use of metformin, intake of often high quantities of non-nutritive sweeteners, inflammation (measured as CRP), and increased gastrointestinal permeability (measured as zonulin), which were variables included in this study, are known causes of faecal dysbiosis [42,43,44,45,46]. In this study, none of these variables were associated with changes in the microbiota.

The alpha diversity was lower in subjects with obesity than in healthy controls, and interestingly, the alpha diversity increased six months after bariatric surgery. Only the species *Alistipes shahii* was significantly different from before to six months after bariatric surgery, corrected for multiple testing. This indicates that the increased diversity is general and not explained by single species alone. The lower alpha diversity in subjects with morbid obesity and the increase after bariatric surgery have been reported in several studies [11,47]. High diversity is associated with a healthy gut microbiota [48].

### Strengths and Limitations

The inclusion of subjects with severe obesity (morbid obesity) and healthy subjects from a population-based survey ensures optimal separation of the groups. The risk that confounding factors and not obesity per se caused the microbial changes was reduced by showing that several variables known to be associated with faecal dysbiosis, such as diabetes and use of metformin and non-nutritive sweeteners, were unrelated to the changes in the microbiota. It is a limitation that other differences in the diet and physical activity were not adjusted for. Adjusting for differences in the diet renders necessary a meticulous registration of the diet during a long period. The food frequency questionnaire used in this study was unfit for this purpose. The risk of type I errors was reduced by correcting for multiple testing. The marked discrimination between the morbidly obese and the healthy group after adjustment for multiple testing encourages new and independent hypothesis-driven studies with more clinical data for verification of the results. The minor changes from before to after surgery might be spontaneous changes unrelated to surgery. Follow-up of the healthy volunteers after six months could have strengthened the study.

The small sample size increased the risk of type II error. This applies primarily to the secondary aims, i.e., the microbiota changes after surgery, differences related to the type of surgery, IBS, diabetes, metformin, and NNS. All the data-driven post hoc analyses need confirmation in prospective hypothesis-driven studies.

The importance of the differences in collecting and handling faeces and extraction of DNA between the groups is uncertain. It is essential to standardise the methods for microbiome analysis to enable reproducibility [49]. However, the biological difference remains the largest source of variation within microbiome studies.

The study analysed faecal bulk luminal samples, which are the commonly used samples. However, samples from other parts of the bowel or the mucosa-associated microbiota could have given other results. The assessment was purely taxonomic. Inclusion of the microbe function is desirable in future research.

## 5. Conclusions

The microbiome profiling of the faecal microbiota based on targeting 104 species showed significant differences in 17 species between subjects with morbid obesity and healthy volunteers. The sensitivity and specificity were 78% in a data-driven post hoc analysis. In subjects with morbid obesity, no differences were seen between subjects with and without IBS or from before to six months after surgery. The findings in this pilot study need follow-up in new hypothesis-driven studies.

## Figures and Tables

**Figure 1 microorganisms-09-00664-f001:**
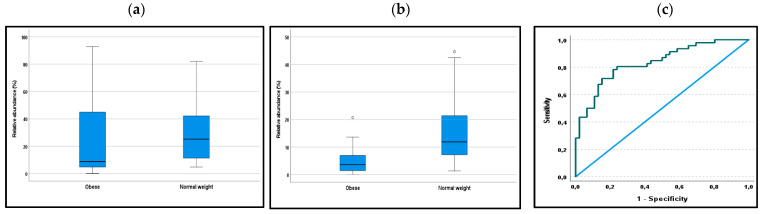
(**a**): The relative abundance (%) of 17 bacteria with significant differences between subjects with morbid obesity and normal weight. (**b**) shows the corresponding results of a post hoc analysis of 8 bacteria that showed the best separation between the groups. The figures show the interquartile ranges, and the circles show two outliers. The differences between the groups were statistically significant (*p* = 0.001 and *p* < 0.001, respectively; Mann–Whitney U-test). (**c**): The receiver operating characteristic (ROC) curve shows the eight selected bacteria’s diagnostic properties for the diagnosis of obesity. The AUC is 0.83 (95% CI: 0.75–0.92; *p* < 0.001).

**Table 1 microorganisms-09-00664-t001:** Participant characteristics.

	Morbidly Obese at Inclusion (no 46)	Healthy Volunteers (no 46)	Morbidly Obese 6 Months after Surgery (no 19)
Age (years, mean (SD))	43.6 (8.5)	43.6 (8.6)	45.7 (6.6)
Gender (men/women; no (%))	2 (4%)/44 (96%)	2 (4%)/44 (96%)	0 (0%)/19 (100%)
BMI kg/m^2^ (mean (SD))	41.9 (3.5)	22.5 (1.5)	30.6 (3.7)
Diabetes (no (%))	4 (9%)	0 (0%)	
Metformin (no (%))	3 (7%)	0 (0%)	
Irritable bowel syndrome (no (%))	23 (50%)	0 (0%)	
IBS-D, IBS-M, IBS-C (no) ^1^	7/11/4		
CRP (mean (SD))	7.0 (5.7)	n.a.	
NNS (units/day) ^2^ (mean (SD))	8.7 (11.9)	n.a.	
Zonulin (mean (SD))	68.3 (37.1)	n.a.	
Operation: Bypass/Sleeve (no (%))			15 (79%)/4 (21%)

^1^ IBS-D: irritable bowel syndrome—diarrhoea predominant, IBS-M: irritable bowel syndrome—mixed, IBS-C: irritable bowel syndrome—constipation predominant. ^2^ NNS = non-nutritive sweeteners. One unit = 100 mL beverage with NNS or two tablets/teaspoons of NNS.

**Table 2 microorganisms-09-00664-t002:** The bacteria showing significant differences between subjects with morbid obesity and normal body weight after adjusting for multiple testing.

Bacterium	Morbid Obesity ^1^	Normal Weight ^1^	Statistics *p*-Value	FDR ^2^ *q*-Value
*Akkermansia muciniphila*	0.02 (0.00–0.44)	0.35 (0.07–3.19)	0.002	0.015
*Anaerobutyricum hallii*	0.00 (0.00–0.00)	0.00 (0.00–0.58)	<0.001	0.001
*Anaerostipes hadrus*	0.03 (0.00–0.36)	0.47 (0.30–1.07)	<0.001	<0.001
*Bifidobacterium adolescentis*	0.00 (0.00–0.46)	0.43 (0.00–1.41)	0.007	0.043
*Bifidobacterium longum*	0.12 (0.00–0.95)	0.72 (0.27–1.52)	0.001	0.008
*Blautia wexlerae*	0.00 (0.00–0.37)	0.55 (0.24–1.32)	<0.001	<0.001
*Butyrivibrio crossotus*	0.00 (0.00–0.00)	0.00 (0.00–0.00)	0.001	0.009
*Christensenella minuta*	0.00 (0.00–0.00)	0.00 (0.00–0.001)	<0.001	0.003
*Coprococcus catus*	0.08 (0.00–0.35)	0.69 (0.38–1.11)	<0.001	<0.001
*Dorea formicigenerans*	0.00 (0.00–0.08)	0.19 (0.13–0.25)	<0.001	<0.001
*Eubacterium siraeum*	0.00 (0.00–0.40)	0.10 (0.03–1.33)	0.005	0.030
*Eubacterium ventriosum*	0.00 (0.00–0.21)	0.27 (0.03–0.52)	0.001	0.009
*Faecalibacterium prausnitzii*	0.81 (0.07–1.85)	1.91 (0.39–3.57)	0.003	0.021
*Haemophilus parainfluenzae*	0.00 (0.00–0.05)	0.06 (0.01–0.16)	<0.001	0.004
*Methanobrevibacter smithii*	0.00 (0.00–0.23)	0.33 (0.00–1.81)	<0.001	<0.001
*Prevotella copri*	0.00 (0.00–10.09)	0.16 (0.00–14.07)	0.002	0.015
*Ruminococcus bromii*	0.07 (0.00–0.88)	2.17 (0.51–4.73)	<0.001	<0.001

^1^ The results are given as the relative abundance (%) of the total number of copies for the given species and reported as median and interquartile range. ^2^ FDR: False discovery rate.

**Table 3 microorganisms-09-00664-t003:** Alpha diversity in groups of subjects reported as median and interquartile range.

Groups	Alpha Diversity	Alpha Diversity	Statistics *p*-Value
Obese/normal weight	Obese 32 (23–36)	Normal weight 42 (37–46)	*p* < 0.001
Surgery	Before surgery 32 (25–36)	After surgery 37 (27–41)	*p* = 0.013
Irritable bowel syndrome	IBS ^1^ Yes 27 (22–35)	IBS ^1^ No 33 (29–37)	n.s. *p* = 0.096
Type of surgery	Gastric bypass 38 (32–41)	Sleeve gastrectomy 32 (22–42)	n.s. *p* = 0.53

^1^ IBS = irritable bowel syndrome.

## Data Availability

The raw datasets generated and analysed during the current study are not publicly available in order to protect participant confidentiality. Case report forms (CRFs) on paper are safely stored. The data were transferred to SPSS for statistical analyses and the data files are stored by Innlandet Hospital Trust, Brumunddal, Norway, on a server dedicated to research. The security follows to the rules given by The Norwegian Data Protection Authority, P.O. Box 8177 Dep. NO-0034 Oslo, Norway. The data are available on request to the authors.

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
