# Peer review of "Are Faecal Microbiota Analyses on Species-Level Suitable Clinical Biomarkers? A Pilot Study in Subjects with Morbid Obesity"

_microorganisms, 2021, doi:10.3390/microorganisms9030664_

Round 1
Reviewer 1 Report
As stated in a previsous reviewing (Microorganisms-1097872) and even if the authors answer the whole comments of the first reviewing, the lack of novelty justify my recommandation, i.e. the manuscript could not be published as the results are extensively studied in the litterature.
As stated by two third of the other reviewers, this manuscript must be rejected.
Finally, note that on a new submission, the track-change mode must be suppressed.
Author Response
Please see the attached letter to the Editor.

Reviewer 2 Report
Dear Authors,
after revisions criticisms remain. Moreover, the research does not add anything new to the field of microbiome and obesity.
Author Response

(The authors gave the same response as above.)

Reviewer 3 Report
After the third review of this manuscript, Changes have been made in the discussion section, including details on limitations of the research method. Only minor issues are of concern.
The primary outcome of this pilot study was the difference in bacterial species levels between obese patients and healthy people of normal weight. If possible, more explanation is needed for the results of the decrease in the relative abundance of 17 bacterial species in obese patients compared to normal weight in the discussion.
Author Response
Please see the attached letter to the Editor.

This manuscript is a resubmission of an earlier submission. The following is a list of the peer review reports and author responses from that submission.
Round 1
Reviewer 1 Report
This article by Per G Farup and Maria G Maseng describe data on GIT/Fecal microbiota using a PCR (Precision Microbiome Profiling) focusing on obesity. Despite this new methodology, the described results have already been extensively studied (and so published).
Some revision are needed in the core manuscript.
Introduction :
- Line 46 : give more details about the 16DNA limitation to genus or species, depending on the method/hypervariable regions used. This is crucial to understand the potential benefit of your method
Method :
- Line 60 : this is a study described in a manuscript, not a "paper".
- Why have the authors decided to apply 2 intervention: diet and surgery?
- Rome III criteria have to be referenced (or even cited).
- Markers (CRP/Zonulin) could be introduced in the introduction more than in the method part, justyfing to limit the blood test to their determination.
Results :
- Table 1 : it is not clear if characteristics of the 19 'patients followed 6 months after surgery" are described separately have also been included in the "morbid obese at inclusion".
- Table 2 : Prefer to indicate median and IQR only per bacteria. Moreover, bacterial names have to be indicated in italic.
- post hoc, et al. has to be indicate in italic (Latin locution)
- Figure 1 is hardly visible and deserve a global title (this remark is similar for other table and figures). please increase their size.
- Line 198 : I think that this "marginal change" has to be more deeply described, as it could be interesting to read.
- Among the 17 bacteria, 14 out of 104 bacteria : please indicate % to ease the reading and understanding.
- Table 3 : prefer Alpha than Alfa
Author Response
Line 46: More details have been given about the 16DNA limitations, lines 46-54.
Line 60: (New line 67) The word “paper” has been replaced by “study.”
The two interventions were according to standard national and international recommendations. This information has been added – lines 96-97.
A reference to the Rome III criteria has been added –, line 114 and reference 26
We prefer to present details like CRP and zonulin in the “Methods” section and not in the “Introduction.”
Table 1: The following sentence has been added, lines 164-165: The 19 participants 6 months after surgery were part of the 46 participants with morbid obesity at inclusion”.
Table 2: Lines 179-180. The results have been changed and reported as median and IQR. The species names have been written with italic font and phylum names in standard font in the tables and the text.
Post hoc has been changed to italics font eight times and et al. three times.
Figure 1. The lack of visibility might be due to the reviewer’s copy. The titles of the figures and tables are according to MDPI’s template.
Line 198. New line 210. The exact result (the AUC with 95%CI) has been given.
The relative abundance of 28, 14 and 17 bacteria has been added the first time the numbers are reported in the paper. Lines 173, 174 and 215.
Table 3. Alfa has been replaced by alpha throughout the paper.
Reviewer 2 Report
In this manuscript, Per and Maria presented the potential of applying Precision Microbiome Profiling, where they quantified 104 gut bacterial species, to the biomarker discovery in the discrimination between morbid obesity and normal individuals. They performed multiple testing correction in the statistical analysis and found 17 out of 104 bacterial species that are reduced in individuals with morbid obesity. In their post hoc investigations, the identified sum of 8 bacterial species provided diagnostic sensitivity and specificity at 78%. Additionally, they observed no microbial difference between individuals with and without IBS, or before and after bariatric surgery, where they assume the lacking in statistical power may derived from limited sample size. The result looks interesting and valuable, however, there are several issues to be addressed before considering for publication.
- The details for post hoc analysis is unclear in the current manuscript. What are the criteria of selecting the eight species to discriminate morbidly obese subjects from normal individuals?
- Consider evaluating the findings from this pilot study in an independent cohort.
- This manuscript lacks physiological or biological interpretation of the identified significantly different bacterial species.
- The details for the reference gene in the qPCR analysis are missing. Normally, primer sequences should be given for reference gene as well as all the targeted genes for individual bacterial species. This is important, without details of qPCR analysis, it will be impossible for independent researcher to reproduce the main findings.
- Table 2 is unreadable, box plots with dots indicating each individual are suggested.
Author Response
The selection of eight species for the best discrimination of subjects with obesity from subjects with normal weight was based on selecting the species with the largest differences between the groups. This has been clarified in the manuscript, lines 188-189.
The need for a new and independent study to verify the results has been added in the “Strengths and limitation” section, lines 319-320.
The findings’ physiological and biological interpretation is outside the scope of this pilot study. The primary aim was to evaluate the method’s ability to separate the faecal microbiota in subjects with morbid obesity from healthy, normal-weight subjects. However, the physiological and biological interpretation is mandatory if the results are confirmed in a new and independent study.
Details for the reference gene in the qPCR analyses. To address this concern additional information about the assays was added to the supplementary table 1, where each species investigated in this manuscript is now linked to an assay number. PMPTM analysis is offered at Bio-Me AS, so thereby it is possible to reproduce the findings by having samples analysed by Bio‑Me with the same PMPTM assays.
Table 2: Lines 179-180. The results have been changed and reported as median and IQR. The species names have been written with italic font and phylum names in standard font in the tables and the text.
Reviewer 3 Report
There is some criticism in the experimental design. First the healthy subjects were not resampled after 6 months nor the subjects without morbid obesity maybe is just a methodological matter, but this offer the opportunity to test if the microbiome is stable or can change during a 6 months window. In any case, the comparison of morbid obesity pre and after treatment is not feasible, also for IBS, considering the different forms of the syndrome (IBS-D, IBS-M, IBS-C) and the low number of subjects for each sub-category, and other. Moreover, for the healthy subjects – or it is not clear– there is not information on dietary habit, which could have affected microbiome composition. Also, there are only 2 and 2 men and the remaining population if female. There are evidences that sex can influence gut microbiome; it is better to recompute the statistics only with female. The only results that can be considered is the comparison of healthy and obese subjects.
Other
Line 247-248: Calculate the ratio e add the values in the text;
Line 248: low? It would be high;
Line 259: which specie?
Line 260-263: It does not seems to be disease specific, considering that the affected subjects were either diabetic, or IBS-D, IBS-M, IBS-C. Please reword.
Line 286-289: Healthy subjects were not analyzed for these biomarkers, why? How can the authors state that none of these variables were associated with changes in the microbiota?
Table 2. The format of the Table is not readable. Report mean (or median) and variations (s.d.) or plot in a box plot.
Table 3. The format of the Table is not readable. Report mean (or median) and variations (s.d.) or plot in a box plot.
Author Response
Reviewer 3 criticises the design, which cannot be changed.
The first comment is the lack of follow up of the healthy volunteers after 6 months because there might be changes in the microbiota in untreated subjects. This limitation has been added to the “Strengths and limitation section”, lines 320-322.
The reviewer states that comparing subjects with obesity before and after surgery is not feasible because half of them have IBS. Comorbidity is prevalent in subjects with morbid obesity and of importance for the microbiota, but the relative changes after surgery are probably unrelated to the comorbidity. No changes have been made in the manuscript.
As stated by the reviewer, dietary habits could have affected the microbiome. The diet likely differs between subjects with obesity and normal weight. The reasons for the differences in the microbiota between the groups have not been part of the study. A comment has been added, lines 316-317.
The reviewer mentions that the skew gender distribution could influence the gut microbiome, which is correct. Since the gender distribution in this matched case-control study is identical in the groups with obesity and normal weight, it is unlikely that the skew gender distribution affects the comparisons between the groups.
The reviewer states that: “The only results that can be considered is the comparison of healthy and obese subjects”. This comparison was the primary aim of the study. However, other comparisons are also of interest, such as in obese subjects with and without IBS. Analyses of subgroups of IBS were not possible due to the small sample size.
Lines 247-8. The Firmicutes/Bacteroidetes ratio was not calculated and reported in the “Results” section. To calculate this ratio study gives no meaning because the analyses included only 104 out of more than 1000 different bacterial species and only a minority of all Firmicutes species. In lines 261-264 in the “Discussion” we wanted to draw the attention to a possible deviation of the ratio.
Line 248. New line 261 - low is correct.
Line 259. New lines 271-272. The reviewer wants a specification of the species mentioned in the sentence “The most predictive species in the study by Meijnikman et al. were not congruent with this study”. To mention a lot of species not congruent with the 17 species in this study seems inappropriate.
Lines 260-3. New lines 272-274. The sentence about disease specificity refers to Gupta et al. and Meijnikman et al. and to reference no 19. The paper discusses that the reason for not finding differences between the diseases (diabetes, IBS etc.) in this study might be a type II error because of the relatively small sample size of the subgroups (lines 323-325). No changes have been made.
Lines 286-9. New lines 297-302.The reviewer asks for the biomarkers: Diabetes, use of metformin, intake of non-nutritive sweeteners, inflammation (measured as CRP) and increased gastrointestinal permeability (measured as zonulin). These variables were not measured in the healthy volunteers, but in all subjects with obesity. The “Result” section gives the comparisons of the microbiota between obese subjects with and without these biomarkers. Therefore, the paper states that none of these variables was associated with changes in the microbiota in subjects with obesity. No changes have been made.
Table 2: Lines 179-180. The results have been changed and reported as median and IQR. The species names have been written with italic font and phylum names in standard font in the tables and the text.
Table 3 has been changed in the same way as table 2.
Reviewer 4 Report
The manuscript by Per G Farup and Maria G Maseng aims to evaluate the diagnostic properties of a new method in 54 subjects with morbid obesity and IBS. The idea is not new and I did not see any significant advancements in this field. More subjects should be involved. As for the bacteria detected in this study, most of them have been reported before. I did not see any new findings in this study. The manuscript is substandard for publication in Microorganisms.
Author Response
The reviewer asks for more participants, which is not possible in this study.
Reviewer 5 Report
The manuscript entitled "Are faecal microbiota analyses on species-level suitable clinical biomarkers? A pilot study in subjects with morbid obesity" showed that differences in intestinal bacteria differ at the species level in patients with severe obesity and healthy subjects of normal weight.
My comments are below.
Although the types of species found vary from study to study, the authors' study of differences in relative abundance at the species level between the obese patients and the control subjects in this case-control study is consistent with results that have already been shown by quite a few studies. In other words, as far as I am aware, the results of a decrease in relative abundance in obese patients are consistent with some previous studies.
This study was not able to compare the clinical significance of fecal microbiota, which differs in relative abundance in the obese group and the normal group, because the clinical data such as CRP, zonulin, etc. are lacking in the control groups (normal weight subjects). It was only presented to see correlation within the group of obese patients, but there was no statistical significance.
In my opinion, the direction of future research will be directed towards confirming comparative studies with at least clinical data and the abundance of microbiota in obese patients and normal control patients.
Author Response
The reviewer has no specific proposals for changes of the paper, but recommends future research with more clinical data directed towards confirming the findings. We agree and have added the need for such studies in line 320.
Round 2
Reviewer 1 Report
Even if the authors answer my whole comments, the lack of novelty justify my recommandation. The manuscript could not be published as the results are extensively studied in the litterature.
Reviewer 3 Report
The manuscript was not improved and as such, the reserach is not published.
Reviewer 4 Report
The authors have addressed my major concerns. I suggest to accept this manuscript.
Reviewer 5 Report
This is a comparative study of the intestinal microflora of normal weight and normal people in obese patients. Although the sex and age were matched, the intestinal microflora varies mainly by diet. In the study, diet was not controlled. Because it is a comparative study of two different groups and those observed at different times. Therefore, there is a high reasonable doubt about the validity of the results.